# The Combined Impacts of ENSO and IOD on Global Seasonal Droughts

Hao Yin [1], Zhiyong Wu [1,2,*], Hayley J. Fowler [3], Stephen Blenkinsop [3], Hai He [1] and Yuan Li [1]

1   College of Hydrology and Water Resources, Hohai University, Nanjing 210098, China
2   Yangtze Institute for Conservation and Development, Hohai University, Nanjing 210098, China
3   School of Engineering, Newcastle University, Newcastle upon Tyne NE1 7RU, UK
*   Correspondence: zywu@hhu.edu.cn

**Abstract:** Previous studies have revealed that global droughts are significantly affected by different types of El Niño–Southern Oscillation (ENSO) events. However, quantifying the temporal and spatial characteristics of global droughts, particularly those occurring during combined ENSO and Indian Ocean Dipole (IOD) events, is still largely unexplored. This study adopts the severity-area-duration (SAD) method to identify large-scale drought events and the Liang-Kleeman Information Flow (LKIF) to demonstrate the cause-and-effect relationship between the Nino3.4/Nino3/Nino4/Dipole Mode Index (DMI) and the global gridded three-month standardized precipitation index (SPI3) during 1951–2020. The five main achievements are as follows: (1) the intensity and coverage of droughts reach a peak in the developing and mature phases of El Niño, while La Niña most influences drought in its mature and decaying phases. (2) Compared with Eastern Pacific (EP) El Niño, the impacts of Central Pacific (CP) El Niño on global drought are more extensive and complex, especially in Africa and South America. (3) The areal extent and intensity of drought are greater in most land areas during the summer and autumn of the combined events. (4) The spatial variabilities in dryness and wetness on land are greater during combined CP El Niño and pIOD events, significantly in China and South America. (5) The quantified causalities from LKIF reveal the driving mechanism of ENSO/IOD on SPI3, supporting the findings above. These results lead to the potential for improving seasonal drought prediction, which is further discussed.

**Keywords:** global droughts; ENSO; IOD; causality analysis

## 1. Introduction

Drought is a hydrometeorological phenomenon that occurs under all climate regimes [1]. From 1998 to 2017, droughts triggered global economic losses of roughly USD 124 billion. Meanwhile, in 2022, more than 2.3 billion people are facing water stress, and almost 160 million children are exposed to severe and prolonged droughts [2]. Although it is unclear whether the coverage and frequency of global droughts have increased significantly over the past decades [3–6], it is expected that when droughts occur now, they are likely to initiate more quickly and become more intense under global warming [5,7,8].

All droughts originate from a precipitation deficit over a prolonged period, with a meteorological drought occurring first [9–11]. Therefore, investigating the mechanisms of the occurrence and evolution of meteorological droughts is important and can provide references for drought prediction to resist droughts. Previous studies have revealed that land-atmosphere interactions, persistent large-scale circulation anomalies or patterns, and large-scale ocean thermal anomalies are the main natural contributors [12]. Land atmospheric interactions/feedbacks favor meteorological droughts by influencing evapotranspiration rates [13–16]. However, these land-atmospheric interactions mostly act as local factors [16,17], while meteorological droughts are primarily controlled by atmospheric circulation and sea surface temperatures [18]. Large-scale circulation anomalies or patterns

can be divided into three types contributing to meteorological droughts. One is the propagation of planetary-scale or large-scale waves in the horizontal direction [19], the second is the vertical descending motion [20], and the third type is the anticyclonic circulation that often drives droughts worldwide [21–24]. According to the final natural contributor, as the strongest large-scale climate modes in the Pacific Ocean and the Indian Ocean, the El Niño–Southern Oscillation (ENSO) and the Indian Ocean Dipole (IOD) significantly impact droughts at the global scale.

Many studies have investigated the associations and interactions between ENSO, IOD, and meteorological droughts. Regarding the impact of ENSO, global anomalies in rainfall during El Niño and La Niña events are evident [25], as shown in Figure 1. Areas in green (yellow) are likely to become wetter (drier) than normal during the indicated months. The dry-trend regions are mostly from 50° S to 50° N, and their locations are consistent with the ENSO-affected drought hotspots identified by Nguyen, Min, and Kim [18] and Christian, et al. [26]. However, it should be noted that the regions and seasons shown in Figure 1 indicate typical but not guaranteed impacts of ENSO. To further analyze how ENSO modulates droughts, researchers prefer to classify El Niño more precisely, based on the event's central location and intensity. Compared to Eastern Pacific (EP) El Niño, Central Pacific (CP) events tend to cause rainfall deficits during southern China autumn [27], southeastern Australia summer [28], and US winter [29], whilst CP El Niño is also less predictable. In addition, CP El Niño has occurred more frequently, and it is expected to be more frequent in the future [30], while EP El Niño has become less common [27,31,32]. Another complex issue is whether a strong El Niño implies extreme drought. Actually, the coverage and intensity of droughts are strengthened in strong versus weak El Niño events in many areas [33], but there exist uncertainties in specific regions. For example, the positive summer Eurasian teleconnection (EU) pattern directly determined the location and intensity of the 2015 extreme drought in northern China under the 2015/16 strong El Niño [34]. In addition, droughts do not occur on land or in the ocean alone in some cases because the global atmospheric and oceanic circulations are coupled. In recent years, studies of land droughts have revealed this process in detail [35,36]. Therefore, it is necessary to quantify the probabilities of drought occurrence globally in the context of ENSO.

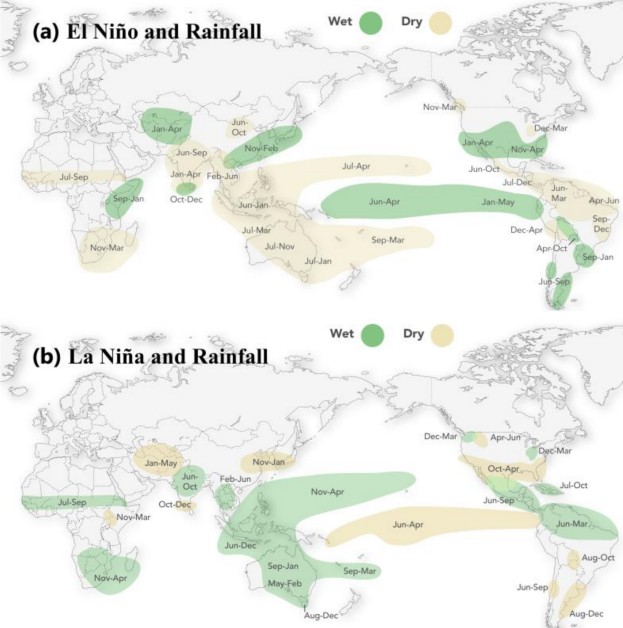

**Figure 1.** Typical rainfall patterns during El Niño and La Niña events based on the whole months of rainfall from 1951–2016 [25].

The interactions between ENSO and IOD can be found in these classic studies [37–40]; we mainly focus on the combined role of these teleconnections in droughts. In 2019, CP El Niño and pIOD resulted in widespread droughts. With the CP El Niño and IOD sea surface temperature (SST) forcings prescribed simultaneously, the experiments run by the Geophysical Fluid Dynamics Laboratory (GFDL) global Atmospheric Model version 2.1 suggested that the superimposed effects of CP- and IOD-related SST anomalies tend to amplify the drought intensity and coverage in Australia [41,42]. Furthermore, CP El Niño and pIOD explained 60% of the intensity and 40% of the amplitude for the 2019 extreme drought in the mid-to-lower reaches of the Yangtze River, which was examined using the NCAR Community Atmosphere Model version 5 [43,44].

In addition to dynamical models, statistical analysis is another method commonly used in climate science. When analyzing the relationship between two variables, correlation analysis is used widely, including Pearson correlation analysis [45], empirical orthogonal function analysis [46], cross-correlation analysis [47], and so on. Correlation analysis can test the relationship between two variables. However, two variables changing together does not mean that one variable causes the other to change, which means a strong correlation may not mean the presence of causality [48]. As a result, we adopt Liang-Kleeman Information Flow (LKIF) as our causality analysis method to quantify the cause and effect between time series [49]. Compared with Granger causality analysis and transfer entropy, LKIF provides quantitative information and dramatically reduces the calculation time [50]. This method has been used to detect the cause-and-effect relation between El Niño and IOD [51]. With this method, we can overcome the problem that traditional statistical methods can only reveal correlations between data. Although global climate models disagree on the ENSO/IOD intensity and frequency in the future [52], it is evident that the frequency of combined El Niño and IOD events has increased since 1965 [53]. Most studies focus on meteorological droughts in a specific year or a fixed region [54–56]. However, as noted previously, there are different classifications for ENSO, and different combinations may imply diverse impacts on droughts globally. Therefore, it is essential to quantify the contribution of ENSO/IOD to global droughts and analyze the changes under combinations of various types of ENSO and IOD.

To better represent climatic consistency and regional climate features, Iturbide et al. [57] presented an updated version of the IPCC climate reference regions for subcontinental analysis (Figure 2). In this study, we investigate the characteristics, evolution, and drivers of meteorological droughts and provide some new insights based on these reference regions. We use the severity-area-duration (SAD) method to identify large-scale drought events, which reflect the effects of ENSO/IOD on global droughts and the drought timing and duration in climate reference regions. Composite analysis and casual interference analysis are then used to quantify the relationships between ENSO/IOD and meteorological droughts. The remainder of this paper is structured as follows: Section 2 introduces the data and methodology. Section 3 presents the results. The conclusions are presented in Section 4.

## 2. Data and Methods

### 2.1. Precipitation and SST Data

The global monthly gridded precipitation data used in this study are from the ECMWF Reanalysis v5 (ERA5, https://www.ecmwf.int/en/forecasts/datasets/reanalysis-datasets/era5, accessed on 30 May 2022) for the period 1950–2020, with a resolution of $0.5° × 0.5°$. Additionally, the SST data are from the monthly mean Hadley Centre Sea Ice and Sea Surface Temperature data set (HadISST1, https://www.metoffice.gov.uk/hadobs/hadisst, accessed on 30 May 2022) for the same period, with a resolution of $1° × 1°$.

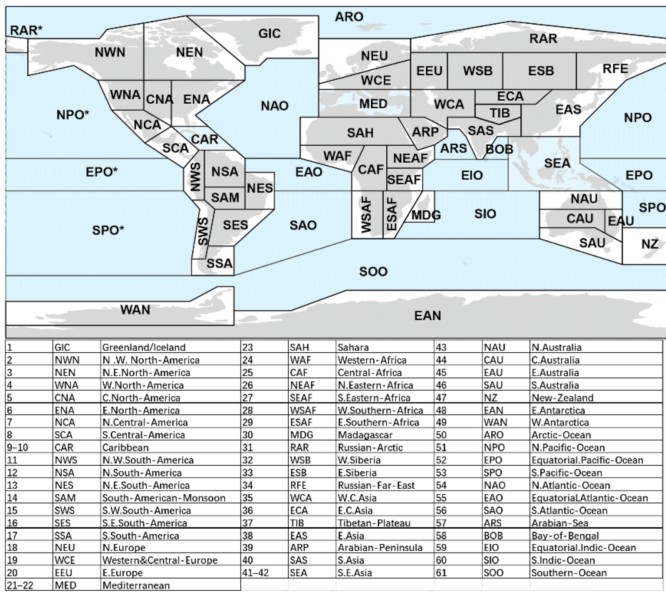

**Figure 2.** Updated IPCC reference land (grey shading) and ocean (blue shading) regions [57].

## 2.2. ENSO and IOD Indices

Based on the HadISST1 data, we calculated the Nino3 (150° W–90° W, 5° S–5° N), Nino4 (160° E–150° W, 5° S–5° N), and Nino3.4 (5° S–5° N, 170°–120° W) indexes. Following the identification standard for ENSO events in Ren, et al. [58], an ENSO event is defined when the absolute value of the 3-month moving average of the NINO3.4 index reaches or exceeds 0.5 °C and lasts for at least 5 months (NINO3.4 index $\geq$ 0.5 °C is an El Niño event, NINO3.4 index $\leq$ −0.5 °C is a La Niña event). An El Niño event with an absolute value of Nino3 (Nino4) index at or above 0.5 °C for at least 3 months is classified as EP (CP) El Niño. The years of different ENSO types (El Niño, EP El Niño, CP El Niño, and La Niña) are shown in Table 1 from 1950 to 2020. In addition, positive and negative IOD events are defined by the Dipole Mode Index (DMI), which is derived from the difference in SST anomalies between the western (10° S–10° N, 50°–70° E) and eastern (10° S–0° N, 90°–110° E) equatorial Indian Ocean. A positive (negative) IOD event is defined in the year when the sliding average of the DMI from September to November is greater (less) than one positive (negative) standard deviation of DMI. Details of the index can be found in Saji et al. [59] and Hameed and Yamagata [60]. The years of all regimes of combined ENSO and IOD events are displayed in Table 2.

**Table 1.** List of El Niño, La Niña, and IOD events during 1950–2020.

| El Niño | | La Niña | IOD | |
|---|---|---|---|---|
| EP El Niño | CP El Niño | | pIOD | nIOD |
| 1951/1952, 1957/1958, 1963/1964, 1965/1966, 1969/1970, 1972/1973, 1976/1977, 1979/1980, 1982/1983, 1986/1987, 1987/1988, 1991/1992, 1997/1998, 2006/2007, 2014/2015, 2015/2016, | 1968/1969, 1977/1978, 1994/1995, 2002/2003, 2004/2005, 2009/2010, 2018/2019, 2019/2020, | 1954/1955, 1955/1956, 1956/1957, 1964/1965, 1970/1971, 1971/1972, 1973/1974, 1975/1976, 1983/1984, 1984/1985, 1988/1989, 1998/1999, 1999/2000, 2007/2008, 2010/2011, 2011/2012, 2017/2018 | 1951, 1961, 1963, 1972, 1982, 1994, 1997, 2002, 2006, 2011, 2015, 2017, 2018,2019 | 1954, 1957, 1958, 1959, 1960, 1996, 1998 |

**Table 2.** All regimes of combined ENSO and IOD events during 1950–2020.

| EP El Niño + pIOD | CP El Niño + pIOD | EP El Niño + nIOD | La Niña + pIOD | La Niña + nIOD |
|---|---|---|---|---|
| 1951/1952, 1963/1964, 1972/1973, 1982/1983, 1997/1998, 2006/2007, 2015/2016 | 1994/1995, 2002/2003, 2018/2019, 2019/2020 | 1957/1958 | 2011/2012, 2017/2018 | 1954/1955, 1998/1999 |

### 2.3. Identification of Large-Scale Drought Events

Since rainfall varies greatly in different regions, the concept of drought varies. In order to assess drought better, the World Meteorological Organization (WMO) recommends using the SPI [61]. In this study, we use SPI3, calculated from monthly ERA5 precipitation data for the whole year, as the seasonal drought index [62–64]. The detailed calculation algorithm for SPI3 can be found in McKee, et al. [65]. Here, the global gridded SPI3 dataset during 1951–2020 is computed via the Climate Indices Python package [66] for convenience.

Using the SPI3 dataset, we perform a global drought analysis from 1951 to 2020. Taking the continuity of time and space into account, we adopt the SAD drought diagnosis method [67–70] to identify large-scale drought events. In contrast with traditional studies, which analyze the intensity, severity, and duration of drought over a fixed region, the SAD method specializes in simultaneously tracking the development of droughts in space and time based on a gridded dataset [71]. The SAD method is briefly outlined as follows.

The SPI3 dataset is three-dimensional (month × latitude × longitude), and we first need to identify the drought grid points on the two-dimensional data at each time step (month). Specifically, we regard a grid with a SPI3 value below −1.0 as being under drought and consider connected areas within which all grids have a SPI3 below −1.0 as a drought cluster. To track clusters through time, we link clusters with overlapping grid cells between time t and time t + 1, while clusters with an area less than 500,000 km$^2$ (often used in global or continental droughts identification [67,69,71]) are removed. Notably, this means that Sahara droughts are not examined in our study. The spatial-temporal evolution of global drought events from August to November 2019 identified using the SAD method is given as an example in Figure 3.

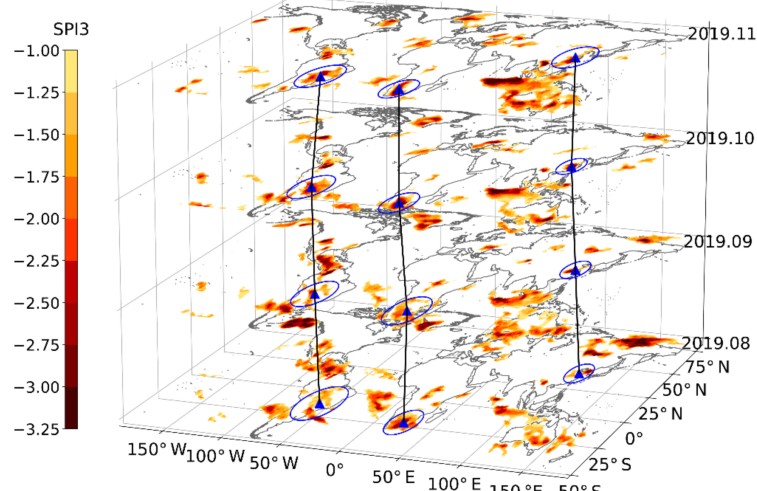

**Figure 3.** Global drought evolution from August to November in 2019. The droughts in China, South Africa, and South America are marked respectively, where the black line represents the evolution of a drought, the blue triangle represents the center of a drought at a time step, and the blue circle covers the drought area roughly.

### 2.4. Causal Analysis between ENSO/IOD and SPI3

As described in Section 2.2, we derived the time series of Nino3.4, DMI, and global SPI3 at all grids for the whole year from 1951 to 2020. LKIF is used to quantify the causality between them. Causality is measured as the time rate of information flow (IF) from one time series to another. It has long been recognized that a non-zero IF, or information transfer as it may appear in some literature, from one event to another logically indicates the strength of the causality from the former to the latter, and a vanishing causality must entail a zero flow [72]. A brief introduction of LKIF is outlined as follows.

Given a two-dimensional stochastic system,

$$\frac{dx_1}{d_t} = F_1(x_1,\ x_2,\ t) + b_{11}w_1 + b_{12}w_2 \tag{1}$$

$$\frac{dx_2}{d_t} = F_2(x_1,\ x_2,\ t) + b_{21}w_1 + b_{22}w_2 \tag{2}$$

where $w_i$ ($i = 1,\ 2$) indicates white noise, $b_{ij}$ and $F_i$ are arbitrary functions of $X$ and $t$.

The rate of information flow from $X_2$ to $X_1$ is:

$$T_{2 \to 1} = -E\left[\frac{1}{\rho_1}\frac{\partial(F_1\rho_1)}{\partial x_1}\right] + \frac{1}{2}E\left[\frac{1}{\rho_1}\frac{\partial(b_{11}^2 + b_{12}^2)\rho_1}{\partial x_2}\right] \tag{3}$$

where $E$ stands for mathematical expectation (units: nats per unit time), and $\rho_1 = \rho_1(x_1)$ is the marginal probability density of $X_1$. If the evolution of $X_1$ is independent of $X_2$, then $T_{2 \to 1} = 0$. The nat is a natural unit of information, based on natural logarithms and powers of e [73]. One nat is the information content of an event when the probability of that event occurring is $1/e$.

When only two time series $X_1$ and $X_2$ are given, the maximum likelihood estimator of Equation (3) is very concise as follows:

$$T_{2 \to 1} = \frac{C_{11}C_{12}C_{2,\ d1} - c_{12}^2 C_{1,\ d1}}{c_{11}^2 C_{22} - C_{11}c_{12}^2} \tag{4}$$

where $C_{ij}$ ($i,\ j = 1,\ 2$) is the sample covariance matrix between time series $X_i$ and $X_j$, and $C_{1,\ dj}$ is the sample covariance between $X_i$ and $X_j = \left\{\frac{X_{j,\ n+1} - X_{j,\ n}}{\Delta t}\right\}$, with $\Delta t$ being the time step size (units: nats per unit time). If $|T_{2 \to 1}|$ is nonzero, $X_2$ is causal to $X_1$; if not, it is non-causal. In this study, the SPI3 at each grid is $X_1$; Nino3.4, Nino3, Nino4, and DMI are $X_2$, respectively, so we can calculate the information flow from Nino3.4/Nino3/Nino4/DMI to SPI3. All confidence intervals reported here are significant at the 95% level, which follows Liang [50,74].

### 2.5. Metrics

It is known that seasonal precipitation variation corresponds to the evolution of ENSO in many regions [75–77]. To quantify the impacts of the different ENSO types, we calculate the drought proportion in each season globally. The equation of drought proportion for each grid is as follows.

$$Proportion = \frac{n}{N} \tag{5}$$

where $N$ represents the total number of months in a season when one ENSO type occurs. For example, there are 17 La Niña years from 1950 to 2020. So $N$ equals $3 \times 17$ when we calculate the drought proportion in JJA, and the $n$ is the total number of months a grid is in drought during the $N$ months.

To match the seasonal scale of SPI3, we process all the atmospheric variables to normalized seasonal anomalies.

$$Normalized\ Seasonal\ Anomalies = \frac{X - \mu}{\sigma} \tag{6}$$

where $X$ represents the original variable, and $\mu$ and $\sigma$ are the climatological mean and standard deviation, respectively. Strictly speaking, $X$ is the 3-month running mean, whose value is located in the last month of the running window. The two climatological variables ($\mu$, $\sigma$) are further computed based on the 3-month running mean.

## 3. Results

### 3.1. Global Drought during ENSO Events

3.1.1. The Proportion of Global Droughts during Different ENSO Types

As illustrated in Figures 4 and 5, meteorological droughts are located in Southeast Asia, Australia, Central America, North South-America, Central Africa, and South Africa during El Niño years, which is consistent with Figure 1 and previous studies. The intensity and coverage of droughts change with the evolution of El Niño and reach their peak during autumn and winter. The distribution of droughts also transforms. For example, El Niño may cause droughts in East Asia and South Asia in the developing and mature phases, while La Niña influences drought most in the mature and decaying phases. As shown in Figure 4f, the percentage of drought coverage is quite high around 15° S and 15° N. The possible cause of this is the descending branch of the anomalous Hadley circulation. In contrast to El Niño years, droughts in East Asia, the Arabian Peninsula, and Central and East Africa are more frequent in La Niña years, and fewer areas of drought are on land.

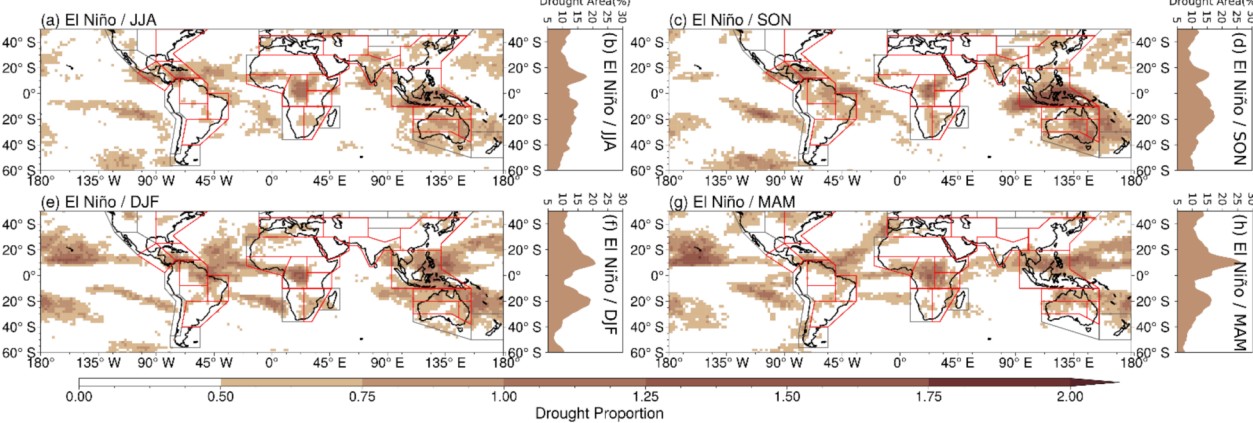

**Figure 4.** The drought proportion in all seasons from 1951–2020 when El Niño events occur. (**a,c,e,g**) represent the drought proportion in Jun-Jul-Aug (JJA), Sep-Oct-Nov (SON), Dec-Jan-Feb (DJF) of the current year, and March-April-May (MAM) of the following year. (**b,d,f,h**) represent the percentage of drought areas at different latitudes. The land regions with red edges are easily affected by ENSO. The easily-affected regions are where the yellow areas in Figure 1 and the areas with high drought proportion in Figure 4 overlap.

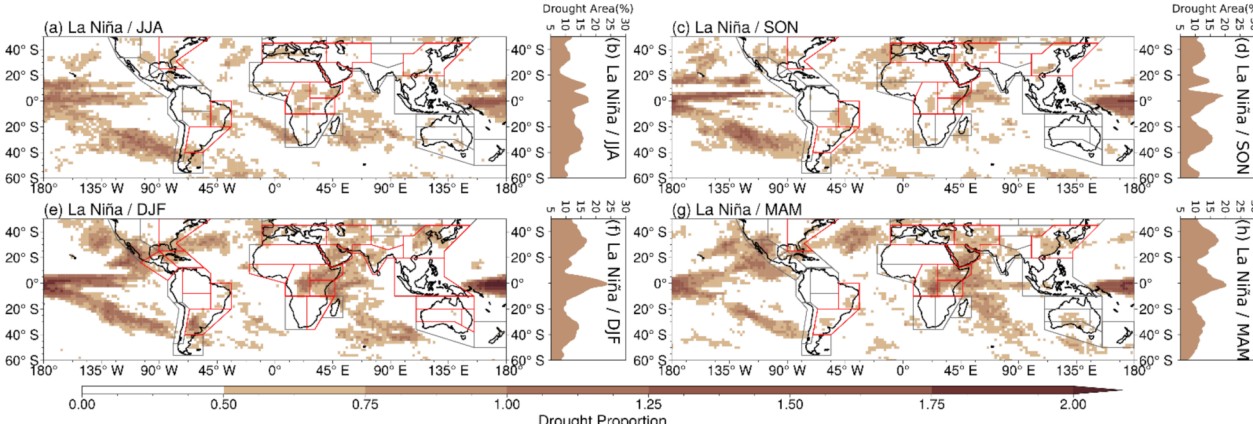

**Figure 5.** The same as Figure 4, but for La Niña.

Figure 6 indicates that the impacts of CP El Niño on global drought are more extensive and complex compared with EP El Niño. In addition to the regions commonly influenced, the occurrence of drought is possible in West North-America and East Asia during the developing phases of CP El Niño. It should be noted that Africa and South America could suffer more intense and widespread droughts under CP El Niño rather than EP events.

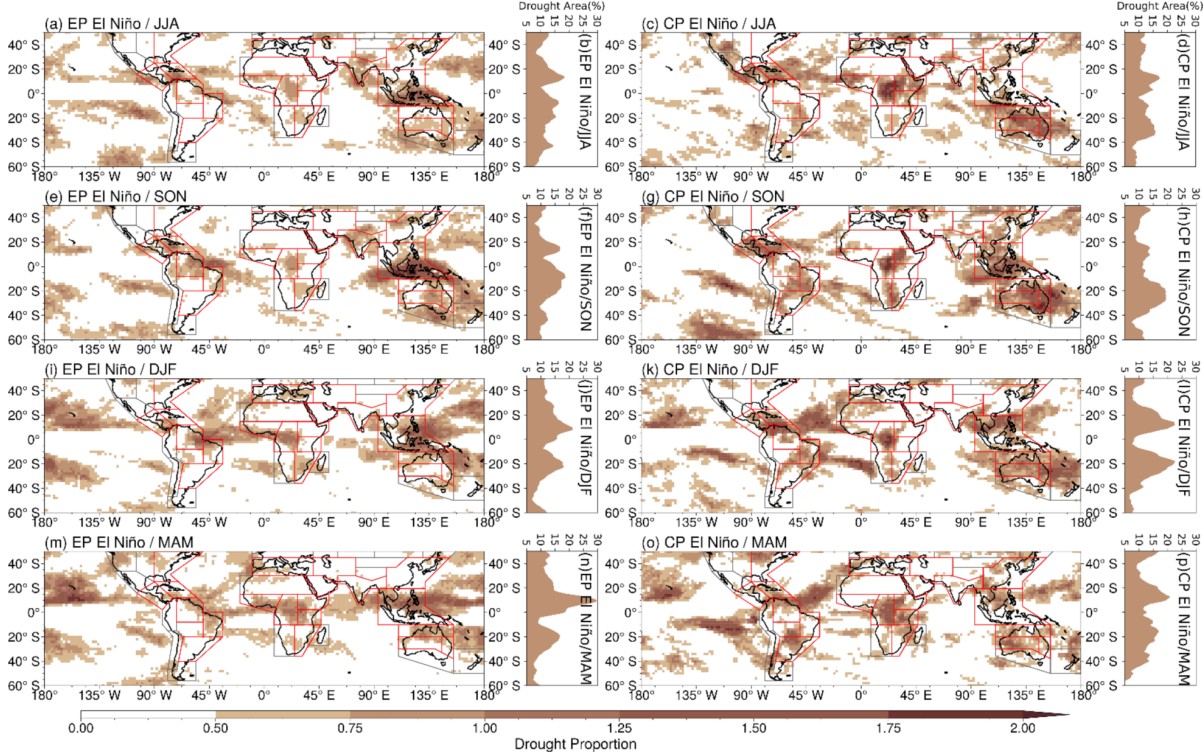

**Figure 6.** The same as Figure 4, but for EP El Niño and CP El Niño.

### 3.1.2. The Significant Drought Timing and Duration in Climate Reference Regions

To quantify the ENSO-induced drought timing and duration in El Niño, CP El Niño, EP El Niño, and La Niña years, we carry out a Student t-test to test the significance of the drought coverage in each month during ENSO years (Figure 7). It should be noted that the results in Figure 7 represent the statistics of global droughts rather than general dryness, in contrast to Figure 1. Although many regions could be dry for long periods in Figure 1, the meteorological droughts in these regions are not significantly influenced by El Niño, and the duration of droughts is short. For example, the duration of the rainfall deficit is from

June to March in North Central-America in Figure 1, whereas the months at a significance level of 90% are from January to March in Figure 7. Compared to La Niña, El Niño affects droughts on land more significantly, consistent with Figure 1.

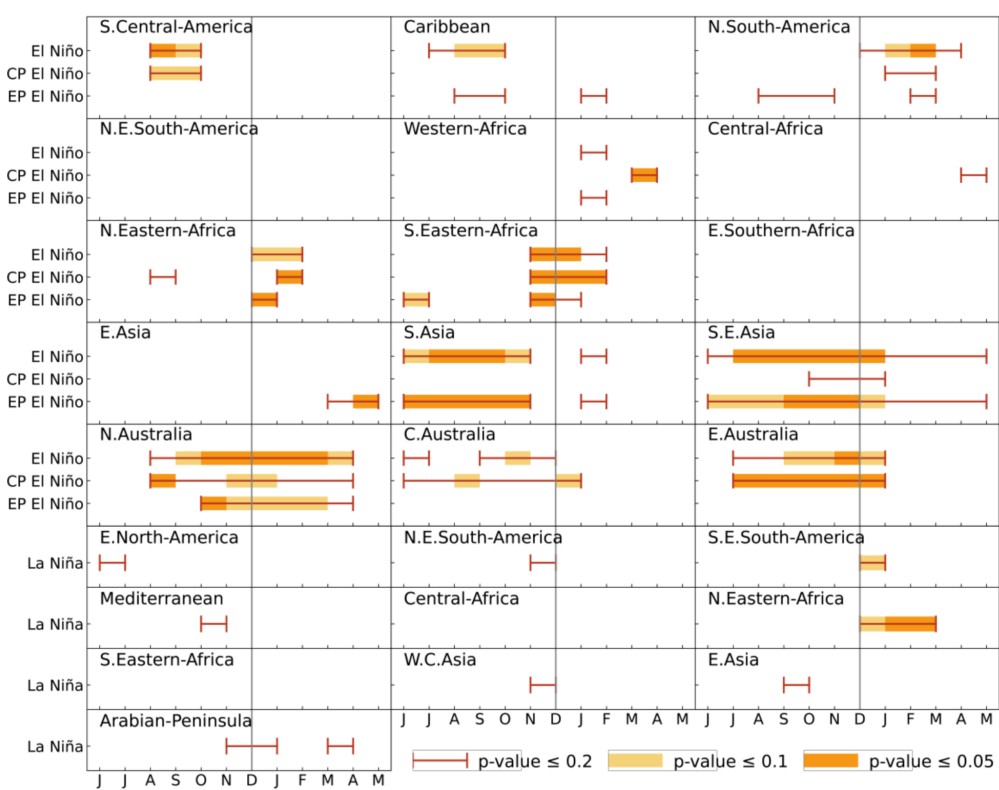

**Figure 7.** Comparisons of ENSO-induced drought timing and duration for SPI3 during El Niño, CP El Niño, EP El Niño, and La Niña. Light and dark filled bars indicate the statistical significance at the 90% and 95% confidence level of drought coverage from June to the next May, respectively. Error bars represent the statistical significance of drought coverage at the 80% confidence level.

In addition to the regions, including South Asia, Southeast Asia, and Australia, vulnerable to El Niño during its evolution, we now pay attention to regions affected in specific phases. Droughts in East Africa are easily detected in the boreal winter, the mature phase of El Niño. However, the significant months of drought in West and Central Africa are much later and shorter. Another interesting finding is that El Niño influences droughts in South Central-America and the Caribbean region during the developing phase, but in South America during the mature and decaying phases.

### 3.2. Global Drought during Combined El Niño and pIOD Events

The occurrences of IOD events can be independent of ENSO or occur simultaneously with ENSO [78]. In this study, we focus on whether combined ENSO and IOD events cause more droughts. Table 2 shows five regimes of combined ENSO and IOD events from 1950 to 2020. Among them, El Niño and pIOD is the most common combination. Seasonal phase locking is an important characteristic of IOD, with significant anomalies appearing around June, intensifying in the following months, and peaking in October [59]. Thus, the changes in droughts in summer and autumn are most interesting.

As displayed in Figures 4 and 5, the drought coverage is greater in autumn than in summer, probably due to the higher intensity of El Niño and pIOD in the former. Except for South Asia and South China, where the frequency of drought is slightly reduced, other land regions are more prone to droughts, especially Southeast Asia, Australia, and Africa in Figure 8. As indicated in Figure 8, compared with the combined EP El Niño and pIOD events, the changes of intensity and areal extent of droughts are greater in the summer

of the years when CP El Niño and pIOD occur simultaneously. In addition, the spatial variabilities of dryness and wetness on land are greater. However, during combined CP El Niño and pIOD events, contrasting drought proportion changes are observed in the Yangtze-Huai River basin and South China. The same phenomenon is also observed in North and Northeast South America.

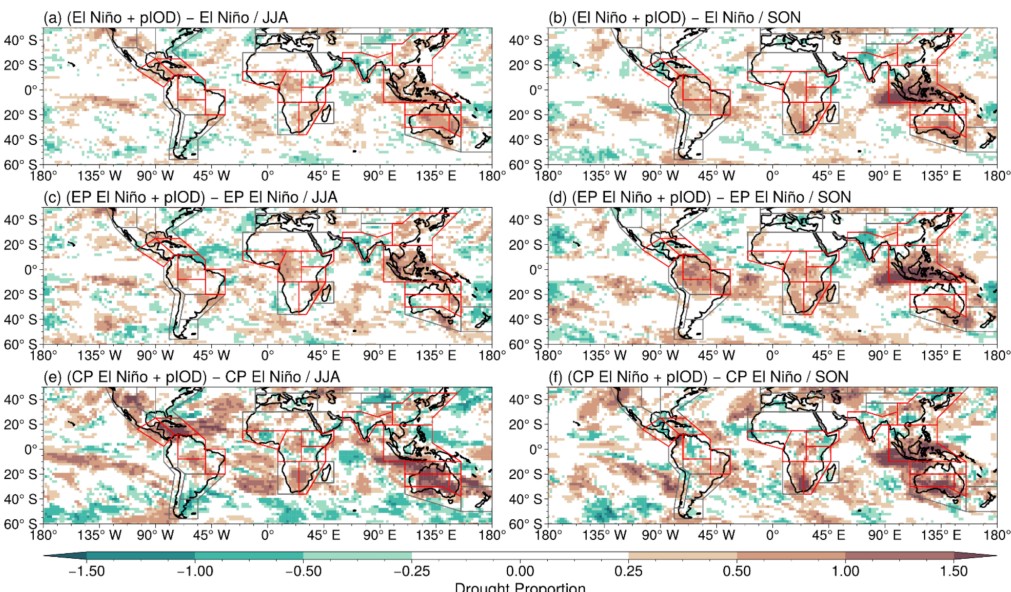

**Figure 8.** The drought proportion in Jun-Jul-Aug (JJA) and Sep-Oct-Nov (SON) when combined El Niño and pIOD occur. (**a**,**b**) is for El Nino, (**c**,**d**) is for EP El Nino, and (**e**,**f**) is for CP El Nino. (El Niño + pIOD) − El Niño represents the difference between the drought proportion when combined events occur and the proportion when only El Niño events occur. The land regions with red edges are easily affected by El Niño.

In order to examine the effects of combined events on drought variability, we also calculate the drought timing and duration (Figure 9). It can be seen that drought is more significant in Southeast Asia and Australia but not in South Asia during combined events. Notably, CP El Niño is more likely to cause drought in Central and East Australia, whether it is an El Niño alone or a combined event. As illustrated in Figure 8, droughts are enhanced across Africa and South America.

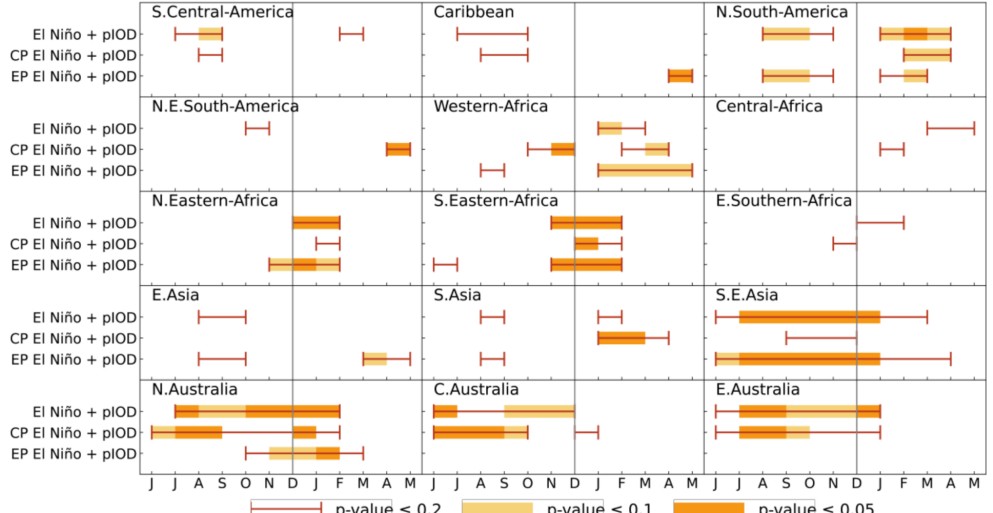

**Figure 9.** The same as Figure 7, but for comparisons during El Niño + pIOD, CP El Niño + pIOD, and EP El Niño + pIOD.

### 3.3. Composite Analysis of Vertical Velocity Anomalies

We have identified that the percentage of drought coverage is highest at around 15° S and 15° N, and at its maximum in winter. As we know, the descending motion of the vertical winds is a direct cause of precipitation deficits, and ENSO events tend to induce anomalies of atmospheric dynamics [79–81]. Therefore, we now perform a composite analysis to investigate the impacts of normalized seasonal anomalies of vertical velocity, as shown in Figures 10 and 11.

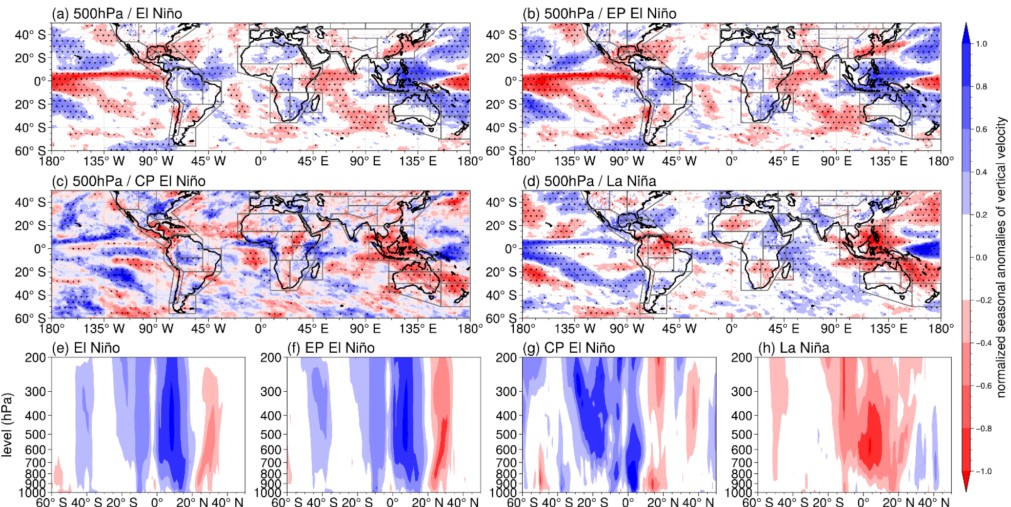

**Figure 10.** Normalized seasonal anomalies of vertical velocity in DJF during (**a**) El Niño, (**b**) EP El Niño, (**c**) CP El Niño and (**d**) La Niña. The red (blue) contours represent vertical descending (upward) motions. (**a–d**) show the vertical velocity anomalies at 500 hPa during these events, and the dotted areas are significant at the 95% confidence level. (**e–f**) are the average of vertical velocity anomalies between 130°–150° E.

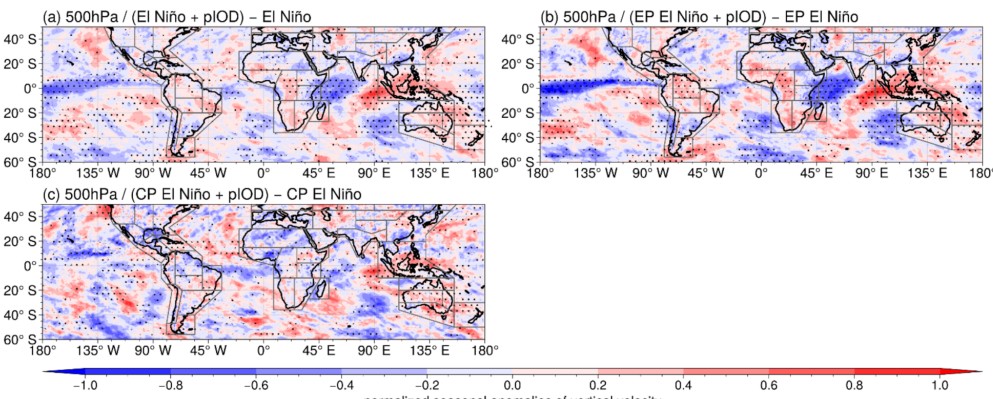

**Figure 11.** Normalized seasonal anomalies of vertical velocity in SON when combined El Niño and pIOD occur. (**a**) is for El Nino, (**b**) is for EP El Nino, and (**c**) is for CP El Nino. (El Niño + pIOD) − El Niño represents the difference between the vertical velocity anomalies when combined events occur and the frequency when only El Niño events occur. The dotted areas are significant at the 95% confidence level.

The red areas in Figure 10a–d almost overlap with the drought areas in Figures 4–6. This suggests that the vertical velocity anomalies at 500 hPa are closely associated with the drought distribution. Previous studies have used vertical velocity to analyze and predict drought [82–84]. Here we examine whether vertical velocity at 500 hPa is sufficient to represent the vertical motion anomalies of the whole atmosphere during drought. In

addition, the spatial distribution of the positive vertical velocity during CP El Niño is significantly more complex than that of EP El Niño, especially in the western Pacific.

Another concern is the latitudinal distribution of the drought area during ENSO events, as illustrated in Figure 10e–h. Compared with CP El Niño, the descending motions are more concentrated and intense near to 15° N in EP El Niño years, resulting in a broader drought coverage than at other latitudes. In contrast, the areas between 30° S to 20° N also experience widespread drought during CP El Niño events.

To further investigate the roles of vertical velocity, Figure 11 presents the differences in normalized seasonal anomalies of vertical velocity for autumn when combined events occur. The red areas in Figure 11 are almost the same as the brown regions in Figure 8, except for North Australia in CP El Niño and pIOD years.

### 3.4. Causal Analysis between Nino3.4/Nino3/Nino4/DMI and SPI3

The quantified causalities of Nino3.4, Nino3, Nino4, and DMI on global SPI3 are displayed in Figure 12. Causalities are significant in almost all the regions where drought proportion increases during ENSO. This demonstrates the mutual validity of our drought identification and causal analysis methods.

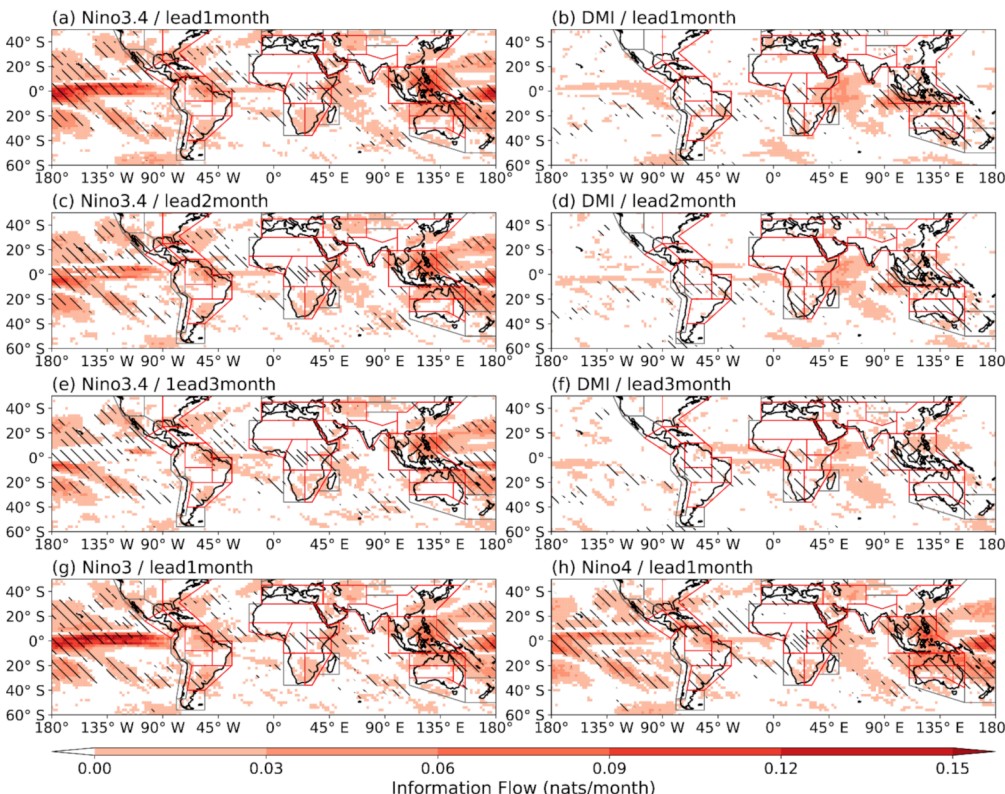

**Figure 12.** Global information flow from Nino3.4, Nino3, Nino4, and DMI to SPI3 from 1951–2020. The colored areas are statistically significant at the 95% confidence level. (**a,c,e**) are the spatial distribution of the information flow between Nino3.4 and the global gridded SPI3 at lead 1, 2, and 3 months, (**b,d,f**) are the spatial distribution of the information flow between the DMI and the gridded global SPI3 at lead 1, 2, and 3 months, and (**g,h**) are the spatial distribution of the information flow between the Nino3/Nino4 and the gridded global SPI3 at lead 1 month. The slashed areas in (**a,c,e**) are the areas of drought coverage over 0.75 in DJF during El Niño and La Niña. The slashed areas in (**b,d,f**) are the areas of drought coverage increase over 0.25 in SON during combined CP El Niño and pIOD. The slashed areas in (**g**) are the areas of drought coverage over 0.75 in DJF during EP El Niño and La Niña. The slashed areas in (**h**) are the areas of drought coverage over 0.75 in DJF during CP El Niño and La Niña.

The information flows of Nino3.4 and DMI decrease as the lead time increases. Compared with DMI, the information flow between Nino3.4 and SPI3 is stronger and wider at all lead months. The regions where the two causal relationships overlap, including Africa and South America, are those where drought frequency is significantly enhanced during combined events. The easily affected regions are all detected with solid information flow, which is consistent with the results in Section 3.2.

However, it is unexpected that there exists causality from South China to Japan, almost coinciding with the trajectory of the atmospheric rivers in East Asia [85,86]. The reason is that the whole series of climate indices from 1951–2020 are used, so Figure 12 reflects the information on dryness and wetness. Although the information flow declines with increasing lead time, the coverage of the information flow from DMI to SPI3 changes. For example, there is no significant causality between DMI and SPI3 at 1 month in East Asia, but the spatial distribution of information flows of DMI and Nino3.4 in the region are similar at lead 3 months. This means DMI may have long-time lag effects on SPI3. This is also revealed in Zhang, et al. [87]; IOD can still exert a stronger influence on precipitation during the ensuing summer.

Since Nino3 reflects the anomalies in the eastern Pacific, the information flow rates in this area and for South America are higher in Figure 12g. In contrast to this, the areas affected significantly by Nino4 are wider in the western Pacific, especially in Australia.

The spatial distribution of the vertical velocity anomalies explains the mechanism behind the changes of drought coverage, and the LKIF reveals the impacts of ENSO on SPI3 by a statistical method. We speculate whether the results from these two completely different approaches are aligned. As indicated in Figure 13, the significant regions of vertical velocity anomalies almost overlap with those of significant causality, except for the Southern Indian Ocean and the Northwest Pacific. Another robust proof is Figure A1, which displays the causal analysis results between ENSO and the vertical velocity anomalies.

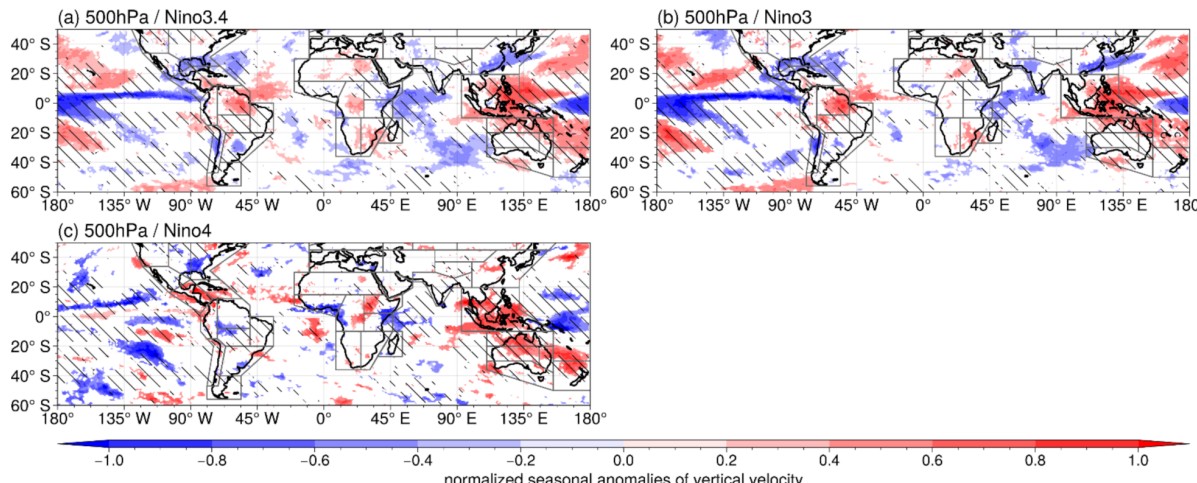

**Figure 13.** Normalized seasonal anomalies of vertical velocity and the information flow from Nino3.4, Nino3, and Nino4 to SPI3 from 1951–2020. The contours and slashed areas are all significant at a 95% confidence level.

## 4. Conclusions

Prior work has documented the roles of ENSO and IOD events in drought causation. Nevertheless, different combinations of ENSO and IOD conditions may imply diverse impacts on droughts globally. This study investigates global drought coverage, frequency, and evolution during ENSO and combined El Niño and pIOD events.

The results suggest that the intensity and coverage of droughts change with the evolution of El Niño and reach their peak during the developing and mature phases, whereas La Niña influences drought most in the mature and decaying phases. Compared with EP El Niño, the impacts of CP El Niño on global droughts are more extensive and

complex, and Africa and South America may suffer from more intense and widespread droughts. We also find that during the summer and autumn of combined El Niño and pIOD events, the total area of drought is greater, and their intensity is enhanced across most land areas. Moreover, the spatial variabilities of dryness and wetness on land are greater during CP El Niño and pIOD events in China and South America. The 500 hPa vertical vorticity anomalies are detected in close association with global droughts, reflecting the dynamic mechanism. Most notably, this is the first study to our knowledge to quantify the information flow from Nino3.4/Nino3/Nino4/DMI to global SPI3 by using LKIF, which reveals the driving mechanism of ENSO/IOD on SPI3, supporting the findings above.

These results provide the potential for improving future seasonal drought prediction. For seasonal drought prediction, ENSO is often used as a predictor related to drought indices. Researchers can construct statistical models between ENSO and seasonal drought [88–90]. Moreover, with the increasing accuracy of ENSO prediction through deep learning and multi-model ensemble projection [91,92], the frequency and intensity of seasonal drought can be estimated based on the mechanisms investigated in this paper.

**Author Contributions:** Conceptualization, H.Y. and Z.W.; methodology, H.Y.; software, H.Y.; validation, H.J.F. and S.B.; formal analysis, H.Y., H.J.F. and S.B.; investigation, H.Y.; resources, Z.W.; data curation, H.Y., H.H. and Y.L.; writing—original draft preparation, H.Y.; writing—review and editing, Z.W., H.J.F. and S.B.; visualization, H.Y.; supervision, Z.W., H.J.F. and S.B.; project administration, Z.W.; funding acquisition, Z.W. All authors have read and agreed to the published version of the manuscript.

**Funding:** This work is supported by the National Natural Science Foundation of China (No. U2240225). The authors gratefully acknowledge the School of Engineering of Newcastle University for providing laboratory and computing resources for use in this study and the funding supported by the China Scholarship Council.

**Institutional Review Board Statement:** Not applicable.

**Informed Consent Statement:** Informed consent was obtained from all subjects involved in the study.

**Conflicts of Interest:** The authors declare no conflict of interest.

**Appendix A**

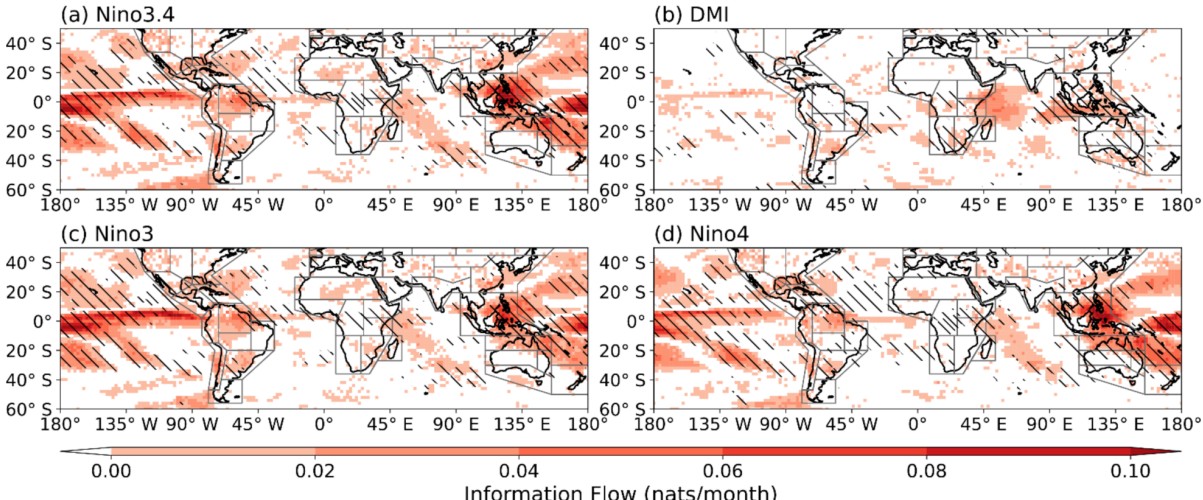

**Figure A1.** Global information flow from Nino3.4, Nino3, Nino4, and DMI to normalized seasonal anomalies of vertical velocity from 1951–2020. The spatial distribution of the information flow is at lead 1 month. The slashed areas are the same as in Figure 12.

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
