# Peer review of "The Combined Impacts of ENSO and IOD on Global Seasonal Droughts"

_atmosphere, doi:10.3390/atmos13101673_

Round 1

Reviewer 1 Report

Combined Impacts of ENSO and IOD on Global Seasonal 2 Droughts

By Hao Yin, Zhiyong Wu, Hayley J. Fowler, Stephen Blenkinsop, Hai He1

and Yuan Li

This manuscript is about global droughts associated with ENSO phases. The drought's extent and duration are examined using different ENSO indexes. This reviewer found that the manuscript is interesting and provides new results. However, my main concern is that the method is not very clear. For example, the months used to analyze the ENSO indexes are unclear. This issue also happens in Table 1,  section 2.2.2. The authors should change the writing to make it clear. The audience can get confused with the present form of the manuscript. Please see my comments below. I strongly recommend checking Section 2 and adding more information that helps to clarify the methods used. Another issue is that the figures are not referred to in the main text. The message "Error! Reference  source not found.." can be found multiple times in the manuscript, making the text difficult to follow.

Major comments

Line 31: Natural disasters do not exist. You can check this webpage from the UN to get more information:

https://www.preventionweb.net/news/there-are-no-natural-disasters

Please remove the word "natural disasters"; authors can use "hydrometeorological phenomenon" instead.

Line 143-144: why do you use a few months? From September to November. Some ENSO events last more than three months.

Table 1. It is not clear which months were used to classify the years. Please add this information.

Section 2.2.2: it is not clear which months are used in SPI3. Was the whole year used? In that case, the created indexes Nino3, Nino4, and Nino3.4 must match the period used for SPI3.

Figure 3: it is better to present several figures in a panel. For the audience, it can be not very clear. I suggest making a panel of 4 figures and marking the regional events with a black box.

Section 2.4: the same comment was made in section 2.2.2. It is not clear how many months the authors use. An ENSO event can last only 6-8 months. How can the authors ensure they are analyzing the same ENSO event?

Line 218: JJA is mentioned, but these months are not used in lines 143-144. I strongly suggest modifying the writing to clarify these points to the readership.

Lines 248-259: It is very difficult to understand the writing because "Error! Reference  source not found.." appears many times in the paragraph.

Minor comments

Line 32: it is unclear which currency the authors use, American dollars? Euros?.

Lines 61-62, 66, 140-141, 174, 220, 227-228, 244-245, 246-247 and so on: This message appears on those lines: Error! Reference  source not found.. Check the reference, please.

Figure 1. It is unclear which months are used to compute the rainfall patterns under different ENSO phases. Please add this information.

Line 180: section 0 does not exist. Please, correct this number.

Line 218-219: please, correct the writing. It is confusing.

Figure 4: I suggest dividing the figure and creating two figures from this one. One figure for El Niño years, and the other one for La Niña years.

Figure 5: similar comment to the last one. I suggest sorting the figure in a different form. The left column can be for EP Niño events, and the right column can be for CP El Niño. Otherwise, it is not easy to compare both definitions and the results.

Figure 10: Why do you use only the period SON for this figure?.

Reviewer 2 Report

-Figure 1 - Aren't the ENSO swapped? Indonesia and Australia should be wet during El-Nino and vice versa.

-Figure 4, 5, A1 - an error in caption "Error! Reference source not found" - also multiple times in text 

Reviewer 3 Report

General Comments:

The paper studied spatial-temporal distribution of global seasonal droughts from statistically-based perspectives (e.g., SAD-based drought detection algorithms, and the introduction of Liang-Kleeman Information Flow), with different joint situations regarding IOD and ENSO considered. In my opinion, the paper is within the scope of the special issue "El Niño-Southern Oscillation Related Extreme Events", and indeed provides some new insight into global-scale ENSO-related drought situations.

In this regard, I show the positive attitude towards this paper, and strongly recommend the publication on this special issue with the following minor comments solved.

Even so, before the formal publication, some implicit descriptions must be improved, and some sections have to be re-organized better. Please try your best to make the paper reader-friendly and quickly understood. I list the relevant comments as follows for the potential consideration. I estimated that the work amount for modification is not heavy and one week is enough. In addition, the following comment is sequentially organized.

Round 2

Reviewer 1 Report

I thank the authors for constructively responding to all my questions. I checked the manuscript, and the main text incorporated the changes. I suggest accepting the manuscript after minor text editing because I still see the error "Error! Reference source not found", for example in line 173. Please, check this issue in the whole text carefully because it also appeared in the first submitted version

Author Response

Daer Reviewer 1,

We appreciate your careful review and time. Your comments and suggestions in two revisions are constructive for improving the paper. We have checked all the references and corrected the errors. 

Thanks a lot again!

Best regards,

Zhiyong Wu